# The Reliability and Validity of the OneStep Smartphone Application for Gait Analysis among Patients Undergoing Rehabilitation for Unilateral Lower Limb Disability

**DOI:** 10.3390/s24113594

**Published:** 2024-06-02

**Authors:** Pnina Marom, Michael Brik, Nirit Agay, Rachel Dankner, Zoya Katzir, Naama Keshet, Dana Doron

**Affiliations:** 1Reuth Research and Development Institute, Reuth Rehabilitation Hospital, Tel Aviv 6772830, Israel; michael.brik@reuth.org.il (M.B.); rachel.dankner@reuth.org.il (R.D.); zoya.katzir@reuth.org.il (Z.K.); 2Department of Health Promotion, School of Public Health, Faculty of Medical & Health Sciences, Tel Aviv University, Tel Aviv 6997801, Israel; 3Unit for Cardiovascular Epidemiology, The Gertner Institute for Epidemiology and Health Policy Research, Sheba Medical Center, Tel Hashomer, Ramat Gan 5262000, Israel; nirita@gertner.health.gov.il; 4Department of Epidemiology and Preventive Medicine, School of Public Health, Faculty of Medical & Health Sciences, Tel Aviv University, Tel Aviv 6997801, Israel; 5Department of General Medicine, School of Medicine, Faculty of Medical & Health Sciences, Tel Aviv University, Tel Aviv 6997801, Israel; 6Department of Physical Therapy, Reuth Rehabilitation Hospital, Tel Aviv 6772830, Israel; naama.keshet@reuth.org.il; 7Ambulatory Day Care, Reuth Rehabilitation Hospital, Tel Aviv 6772830, Israel

**Keywords:** gait analysis, gait dysfunction, smartphone application, reliability, validity, rehabilitation

## Abstract

An easy-to-use and reliable tool is essential for gait assessment of people with gait pathologies. This study aimed to assess the reliability and validity of the OneStep smartphone application compared to the C-Mill-VR+ treadmill (Motek, Nederlands), among patients undergoing rehabilitation for unilateral lower extremity disability. Spatiotemporal gait parameters were extracted from the treadmill and from two smartphones, one on each leg. Inter-device reliability was evaluated using Pearson correlation, intra-cluster correlation coefficient (ICC), and Cohen’s d, comparing the application’s readings from the two phones. Validity was assessed by comparing readings from each phone to the treadmill. Twenty-eight patients completed the study; the median age was 45.5 years, and 61% were males. The ICC between the phones showed a high correlation (r = 0.89–1) and good-to-excellent reliability (ICC range, 0.77–1) for all the gait parameters examined. The correlations between the phones and the treadmill were mostly above 0.8. The ICC between each phone and the treadmill demonstrated moderate-to-excellent validity for all the gait parameters (range, 0.58–1). Only ‘step length of the impaired leg’ showed poor-to-good validity (range, 0.37–0.84). Cohen’s d effect size was small (d < 0.5) for all the parameters. The studied application demonstrated good reliability and validity for spatiotemporal gait assessment in patients with unilateral lower limb disability.

## 1. Introduction

Gait analysis serves as an important tool in clinical settings for identifying gait disorders due to neurological and orthopedic conditions. Quantitative gait analysis also serves as an indicator for clinicians to assess gait patterns related to various functional limitations, to guide clinical decisions, to personalize treatment, and to monitor the effectiveness of interventions [1,2,3].

Human gait can be characterized according to numerous parameters, which are generally classified into five types: spatiotemporal, kinetic, kinematic, electromyographic, and integrated biomechanical. Some spatiotemporal parameters, which include gait speed, stride length, cadence, step width, single and double limb support time, and stance duration, are the most applicable in clinical practice [2,4]. Numerous studies have investigated the evaluation of spatiotemporal parameters in rehabilitation settings, particularly focusing on patients recovering from stroke [5,6,7] and those suffering from chronic pain syndromes [8,9].

The current gold standards for quantitative gait analyses are optical motion-capture systems, force plates, and instrumented walkways [1,2,10]. However, some major drawbacks of these devices are their high costs and time-consuming set-ups, and the necessity of highly trained personnel [11]. Most importantly, these devices are not suitable for examining gait characteristics in peoples’ own natural environment, hence limiting their accessibility and feasibility in both clinical practice and research [12,13,14]. An additional limitation is that they can only measure up to a few steps in non-laboratory walking conditions [15]. Parameters such as gait variability are more reliable when assessed for walking distances of at least 20 m [16,17]. Hence, many of the above-mentioned devices may not represent longer duration gait ability [15]. To overcome this limitation, walking should be assessed outside the laboratory or on treadmills, where individuals can walk for extended periods of time.

The remarkable development of wearable sensor technologies has provided an inexpensive, portable, easy-to-use, and yet reliable and efficient alternative for comprehensive gait analysis [16,18,19,20,21]. Such wearable devices usually rely on wireless body-fixed sensors that include inertial measurement units (IMUs) [22,23]. An IMU is typically an electromechanical or solid-state device that contains an array of sensors that are able to measure motion. That is, to detect a body’s specific force and angular rate using a combination of accelerometers, gyroscopes, and sometimes magnetometers [23]. A range of studies have explored the use of wireless body-fixed sensors with IMUs for gait assessment. These studies collectively highlight the potential of this technology for gait analysis, particularly in terms of precision, usability, and patient monitoring [24,25,26,27].

Today, smartphones are widespread and are standardly embodied with IMUs that enable attaining spatiotemporal gait information. This technology has several potential applications including gait monitoring in different health and therapy settings [28], fall risk assessment [29], and sports and fitness tracking [30]. Smartphone-based assessment has demonstrated reliability and validity in performing spatiotemporal gait analysis among healthy young adults [31,32,33,34,35] and older adults [36], and among populations with various gait pathologies. These include, but are not limited to, Parkinson’s disease [37,38,39], rheumatoid arthritis [40], stroke [41], and musculoskeletal disorders [42,43]. The primary benefits of smartphone-based gait analysis are its affordability, portability, and broad accessibility, unlike specialized gait laboratories [44]. Notably, several investigators have cautioned that gait parameter evaluation may entail inadequate reliability and compromised accuracy [33,34,35,40]. This may be due to the limited number of analyzed steps, the execution of fewer than the 50 recommended steps suggested by Galna et al. [45], slow gait speeds, short strides, and gait asymmetries that are common in individuals with severe pathological gaits [46].

The OneStep smartphone application (Celloscope Ltd., Tel Aviv, Israel) was previously found to be both reliable and valid among healthy persons [33,34] and among people with mild gait-related pathologies [42]. Those studies utilized various reference systems, including an APDM mobility lab (APDM Inc.; Portland, OR, USA) [33], a 10-camera motion analysis system (Vicon Motion Systems; Oxford, UK) [34], and a pressure-sensor gait mat (The ProtoKinetics Zeno™ Walkway, Havertown, PA, USA) [42]. The purpose of the current study was to further assess the validity and reliability of this application, and specifically among those undergoing rehabilitation for unilateral lower limb disability. We compared the spatiotemporal gait parameters between the OneStep smartphone application and the C-Mill instrumented treadmill’s built-in gait tracking system (Motek Ltd., Amsterdam, The Netherlands).

## 2. Materials and Methods

### 2.1. Setting and Study Population

This study took place at Reuth Rehabilitation Hospital in Tel-Aviv, Israel, and included both chronic and acute care patients admitted to the hospital for rehabilitation treatment. All eligible patients aged 18 to 65 years with unilateral lower limb disability as confirmed by their electronic health records (EHR) were offered to participate. Although diagnosed with various conditions, all patients suffered from disability in one leg. The impaired leg was selected based on the patient’s reported symptoms and following consultation with the attending physician (with the other leg being considered as the opposite leg). Additionally, two validated questionnaires, the Lower Extremity Functional Scale (LEFS) [47] and the Brief Pain Inventory (BPI) [48], were used to assess the patients’ disability. The LEFS is presented as a sum of the twenty items; the lower the score, the greater the disability (score 0–80). The BPI is presented by two main areas it assesses: pain severity (the mean score of the four severity items) and pain interference (the mean score of the seven interference items); the higher the score, the greater the pain/disability (score 0–10). Study inclusion criteria were the ability to complete a walking test with or without a walking aid, and a Mini–Mental State Examination (MMSE) cognitive score of 24 or higher. Study exclusion criteria were an existing cardio-pulmonary disease, a history of falls, visual disturbances (not rectified by glasses), pregnancy, or wearing prosthetics or orthoses. Patients unable to walk on the treadmill without walking aids were also excluded; however, the use of handrails was permitted. The final sample size consisted of 28 participants.

This study was conducted according to the guidelines of the Declaration of Helsinki, and approved by the Institutional Review Board (Protocol No: 0003-21-RRH, date of approval: 10 January 2021). This study was registered at ClinicalTrials.gov (NIH Identifier: NCT05009303, date of registration: 17 August 2021).

### 2.2. Study Procedures

Eligible patients who signed an informed consent form were asked to participate in a walking session. They were advised to wear comfortable shoes and pants with front pockets. For those unfamiliar with a treadmill, a short (up to two minute) walk on the treadmill preceded the experiment, with a 10-min break to ensure that the performance during the experimental procedures would not be impaired. During this short walk, the participants chose the optimal treadmill speed, which they maintained throughout the experiment. The participants were equipped with two smartphones, preinstalled with the app, and placed one in each of their pants’ front pockets. The pockets were emptied beforehand. For the purpose of this study, a single type of android smartphone was used (Xiaomi Redmi Note 8, Beijing, China). The two phones were paired before each walking session. The participants were then instructed to walk on the treadmill with the phones in their pockets, for at least 2 min and up to 15 min. They were asked to walk at their own comfortable pace with the slope set to 0 degrees. During the entire test procedures, the instructors, who were licensed physical therapists, were present, to ensure the participants’ safety and to address their needs.

### 2.3. Study Tools and Devices

The OneStep smartphone application (version 2.13.19; Celloscope Ltd., Tel-Aviv, Israel) uses the smartphone’s built-in IMU to measure acceleration and orientation in three dimensions at a sampling rate of 100 Hz. This recorded motion is converted into a digital representation of the walk by OneStep’s proprietary algorithms. Machine learning models then extract a variety of gait parameters from that representation.C-Mill VR+ (Motek, Amsterdam, the Netherlands) is an instrumented treadmill used as a standard reference in this study. The C-Mill is equipped with CueFors^®^ software (version no. 2) to control and track various gait parameters and automatically generate reports for patient evaluation. It is also equipped with a large embedded force platform to quantify center-of-pressure movements (velocity, displacement, sway area, variability, and root mean square displacement). The treadmill records the vertical force and the position of the center-of-pressure at a sampling rate of 500 Hz. The C-Mill treadmill system is ISO 13485 [49] certified and extensively used in both clinical and research settings, thus enabling valid, objective, and highly reliable assessment of balance, gait, and gait adaptability [50]. Three studies have highlighted the use of the C-Mill treadmill for gait assessment in a rehabilitation setting [50,51,52]. Notably, in this study, the C-Mill VR+ treadmill, which is considered the most advanced model of C-Mill, was used; however, the VR system was not activated during this study’s session. Additionally, a variety of studies investigating different treadmill-based systems for gait analysis have consistently shown that an instrumented treadmill is an effective and valid tool for analyzing the spatiotemporal parameters of walking [53,54,55,56]. In addition, treadmills provide a controlled environment where variables such as incline can be precisely adjusted and consistently maintained. This level of control is essential for replicating and standardizing test conditions, ensuring the reliability of our data across different sessions and subjects. Furthermore, the use of a treadmill—a device with which our patients are already familiar due to their rehabilitation treatments—enables us to observe their natural walking patterns in a setting that is both comfortable and familiar to them. This familiarity helps minimize variability that might otherwise arise from patients needing to adapt to unfamiliar equipment, thereby enhancing the authenticity of our data.

### 2.4. Gait Analysis Methods and Data Collection

An outline of OneStep’s gait analysis method: The motion representation process that converts the raw signals from the IMU into a standard representation of the user’s movement operates as follows (as described in patent application US20200289027A1): First, the raw signal is segmented into gait cycles. Then, each cycle’s accelerations are integrated into the sensor’s displacement over the cycle, considering the sensor’s orientation in space. This produces the sensor’s trajectory over each gait cycle in an inertial reference frame that moves in the mean speed of the sensor and thus constrains it to obtain a closed-form trajectory containing the six degrees of freedom (6DOF) of the sensor over the gait cycle. This process is reversible so that all information in the original signal remains in the represented data. In addition, it avoids drifts in integrating accelerations, which is a known issue in inertial data processing. Last, the represented trajectory is rotated around the vertical axis to match the most significant change in the horizontal plane displacement in order to be invariant to the device’s orientation over the end-user’s body. Eventually, this process yields a representation of the 6DOF over each gait cycle, depending only on the user’s movement and the device’s bodily position. It is divided into 100 units in time to provide a standard representation with a 6 × 100 matrix-form.

Deep learning models then obtain this standard-formed representation of movement and predict values for various gait parameters, including spatiotemporal parameters, for each individual gait cycle. These models are based on regression models using convolutional layers, which were trained on labeled data of gait parameter values and the 6DOF representation of their corresponding cycle’s movement.

Data collection: Three instruments simultaneously captured characteristics of the participants’ spatiotemporal gait: the app on the phone placed on the impaired leg, the app on the phone placed on the opposite leg, and the treadmill (Figure 1). Four general and acceptable gait parameters [2,4] were collected: cadence (steps per minute), gait speed (meters per second), stride length (cm), and double limb support (% stance phase). In addition, four gait parameters were collected separately for each leg (impaired and opposite): step length (cm), swing phase (% gait cycle), stance phase (% gait cycle), and single limb support (% stance phase).

Each participant was identified in both the app and the treadmill, only by means of a unique identification number. Gait parameters collected with the OneStep application were uploaded to a cloud server. All the data derived from the instrumented treadmill were collected by the hospital’s study team and provided to the company developing the application. The data were then synchronized from the three devices according to the participants’ unique identification number.

### 2.5. Statistical Analysis

Statistical analyses were carried out in SPSS Version 24 (IBM Corporation, Amonk, NY, USA) and in Excel via the Real Statistics Resource Pack software (Release 7.6). For each gait parameter, the means and standard deviations of measurements from each of the three assessment tools were computed. Subsequently, for each parameter, three comparisons were conducted: between the two phones (one placed on the impaired leg and one on the opposite leg), and between each phone and the treadmill reading. Inter-device reliability was assessed using a number of measurements that compared the app readings of the two phones. First, correlations between each pair of tools were assessed by calculating the Pearson correlation coefficient. Second, to assess the agreement between each pair, the mean difference, Cohen’s d, and the *p*-value for a test for difference were calculated. The Wilcoxon signed-rank test was used, due to the small sample size. Third, the intra-class correlation coefficient (ICC) and its 95% confidence interval were calculated using a two-way mixed-effect linear model (ICC (3,1)). These same measures were used to assess the validity of the comparisons of the treadmill readings with the readings from each phone.

ICC is a widely used index in interrater reliability analyses [57]. In this study, ICC was used to assess inter-device reliability, to measure the consistency between the two phones, and to evaluate the validity of the OneStep application by measuring its consistency with the readings of a standard treadmill. Following the suggested rule of thumb of Koo et al. [57], an ICC above 0.9 was considered excellent, 0.75–0.9 as good, 0.5–0.75 as moderate, and below 0.5 as poor. Bland–Altman plots were used to depict the level of agreement between measurements of the treadmill versus each phone. The tests were two-tailed, and a *p*-value of *p* ≤ 0.05 was considered significant.

Using the sample size determination method outlined by Bonett [58], and with the assumption of ICC being 0.80, this sample size of 28 participants and two raters provides a 95% confidence interval around an ICC of width 0.2 (ICC ± 0.1). This level of precision is deemed satisfactory for our research objectives, indicating that both the sample size and the number of raters are sufficient.

## 3. Results

### Participants and Gait Characteristics

The final sample size consisted of 28 participants, aged 22 to 69 years with a median age of 45.5 years. The majority (61%) were males. The body mass index (BMI) of 13 (46.5%) participants was within the normal range (18–25 kg/m^2^); two (7%) had a BMI under 18 kg/m^2^, and the remaining 13 (46.5%) had a BMI above 25 kg/m^2^. For 16 (57%), the left leg was impaired. Mean (SD) reported LEFS was 30.3 (11.4). Fourteen (50%) patients were diagnosed with single or multiple fractures to the lower limb and 8 (29%) with hemiparesis due to stroke or traumatic brain injury. For the remaining six patients, the ability to walk was impaired due to one of the following diagnoses: chronic pain syndrome, Complex Regional Pain Syndrome (CRPS), or fibromyalgia. The characteristics of the study participants are presented in Table 1.

The walking sessions lasted on average 8.4 min (median 7.9). The median distance covered was 268 m (interquartile range: 182–410) and the median number of steps was 645 (interquartile range: 471–867). The means and standard deviations of selected gait parameters, as measured by each of the three assessment tools, are presented in Table 2.

Table 3 depicts the pairwise comparisons between the readings of the two smartphones, one placed on the impaired leg and one on the opposite leg. As expected, correlations between the two phones were high (r = 0.89–1). The ICC for consistency between the two phones was excellent for cadence and gait speed (range, 0.98–1), and good-to-excellent for all the other parameters (range, 0.77–0.96). The ICC did not differ according to the phone’s location (on the impaired or on the opposite leg). Cohen’s d statistic was trivial or small (under 0.5) for all the parameters.

Table 4 describes the pairwise comparison between the readings of the app (placed on the impaired leg or the opposite leg) and the readings obtained from the treadmill. Correlations between the two phones and the treadmill ranged from 0.65 to 1, but were mostly above 0.8. The ICC for consistency between the phone and the treadmill was excellent for cadence and gait speed (range, 0.92–1) and moderate-to-excellent for all the other spatiotemporal parameters (range, 0.58–0.93), except ‘step length of the impaired leg’. This parameter showed poor-to-good validity (range, 0.37–0.84).

Statistically significant differences were observed between the readings of the treadmill and the smartphone placed on the opposite leg. Statistically significant differences were found between the readings of each phone and of the treadmill in the ‘step length of the opposite leg’ parameter.

The results from the Bland–Altman plots indicated excellent agreement between the gait parameters computed by the app and the instrumented treadmill; the differences were well within the 95% limits of agreement (±1.96 standard deviation of the difference) (Figure 2). The distribution demonstrated no apparent bias between the measures, with no systematic overestimation or underestimation of gait parameters by the app. Further inspection of the plots showed some differences between measurements on the impaired and the opposite leg; however, these were very subtle. Thus, these plots did not indicate a preference for the phone placement on either the impaired or the other leg.

## 4. Discussion

This is the fourth study conducted to assess the validity and reliability of the OneStep smartphone application. The previous studies were conducted among healthy individuals [33,34] or among patients with minor gait-related problems [42], and found the app to be both reliable and valid. Our sample consisted of patients undergoing rehabilitation, with mild-to-moderate lower limb functional disability, with one impaired leg.

The ICC for consistency between the phones showed good-to-excellent reliability (ICC range, 0.77 to 1) and high correlation (r = 0.89 to 1) for all the gait parameters. The lack of statistically significant mean differences between the phones for all the gait parameters and a trivial-to-small (d < 0.5) Cohen’s d effect size are indicators of the high inter-device reliability of the app.

The Pearson correlations of the readings between the phones and the treadmill were mostly above 0.8, indicating the high validity of the app. Some statistically significant differences were found between the readings of the treadmill and the smartphone placed on the opposite leg, but not on the impaired leg. This finding suggests that gait assessment conducted using the phone placed on the impaired leg is more accurate than with the phone placed on the opposite leg. However, the Cohen d‘s effect size was small and the ICC values were moderate-to-excellent for all the gait parameters (with the exception of ‘step length of the impaired leg’). Moreover, agreement was excellent between the app and the treadmill for all the gait parameters, as demonstrated in the Bland–Altman plots, and as presented separately for each foot. These findings suggest that the differences between the readings of the treadmill and the smartphone placed on the opposite leg are negligible, despite the statistical significance.

In light of the above, the findings of the current analysis suggest that the examined smartphone application is both reliable and valid for assessing spatiotemporal gait parameters in patients undergoing rehabilitation for mild-to-moderate unilateral lower limb disability or chronic pain. These findings are consistent with studies that demonstrated the reliability and validity of smartphone-based assessments in performing gait analysis in various populations [31,32,35,36,37,38,39,40,41,59].

Our findings have important clinical implications. First, we were able to demonstrate that the smartphone placement, whether on the impaired leg or the other leg, does not significantly alter the accuracy of the assessment of spatiotemporal gait parameters. The gold-standard gait analysis methods undoubtedly can provide far more detailed information about patients’ gait [1,2,10]. However, such assessment methods are constrained to specialized motion analysis laboratories, which may not accurately reflect an individual’s free-living gait [12,13,14]. The use of a smartphone for gait analysis has many advantages. These include simplicity, operation without calibration or particular skills, high accessibility, and the possibility of collecting valuable clinical data anywhere and at any time. A smartphone-based app for gait analysis can be used in the clinic and can support remote patient monitoring, personalized treatment, and improved continuity of care. Moreover, it can enable large-scale assessments that are not feasible with conventional methods [44]. Second, although not a primary aim of this study, the findings indicate that among patients with a functional disability in one leg—who consequently naturally exhibit slower gait speeds and shorter strides—these characteristics might not affect the app’s accuracy in assessing spatiotemporal gait parameters. Future research should further explore this particular aspect. This study has several limitations. First, the app was installed on a single type of smartphone (Xiaomi Redmi Note 8, China). Two smartphones of the same type, both with a preinstalled app, were provided to all the study participants (some did not have smartphones and others were reluctant to install the app on their own phones). This suggests that other types of smartphones may provide different results. Second, gait analysis using an instrumented treadmill is rather uncommon. Limitations that have been noted are its lower reliability of force records [60] and its lack of an absolute representation of walking in a natural environment; and the biomechanical differences in walking kinematics and dynamics [61]. In contrast, the optical motion-capture systems are considered by many as a gold standard for examining gait parameters [1,2,10], albeit these instruments are not abundant, especially in rehabilitation hospitals. As previous studies conducted to validate the OneStep app used various reference systems including a number of types of smartphones and motion-capture systems, the use of a single type of smartphone and an instrumented treadmill as a reference system in the current study was deemed adequate. Third, all the participants were asked to walk on the treadmill while its slope was set to zero degrees. Therefore, the app’s reliability and validity with other slope settings should be examined, to better resemble walking in various environments. Notably, the slope was intentionally set to 0 degrees as the walking abilities of the patients were limited. Moreover, the walking speed was constant throughout the sessions (as requested by the participants). This reduces this study’s generalizability as walking was not simulated in a natural environment or while doing daily activities. Fourth, while the sample size was sufficient, it was relatively small; however, the study featured a mix of participant demographics (age, sex, and BMI) and a range of causes for unilateral limb disability. Finally, this study was not limited to a specific patient group but covered a diverse patient cohort with neurological and orthopedic conditions to thoroughly assess the application’s validity and reliability. However, all of this study’s participants had disability in one of their legs and were actively engaged in ambulation. Since this study focuses on functional gait aspects rather than specific pathologies, this inclusive approach broadens the generalizability of this study’s outcomes, thereby enhancing the application’s utility in a rehabilitation setting.

## 5. Conclusions

The primary goal of this study was to assess the reliability and validity of a technology that could be easily applied in a rehabilitation hospital among patients, with mild-to-moderate unilateral lower limb disability. The findings suggest that the OneStep smartphone application is a reliable and valid instrument for spatiotemporal gait assessment for this population. The findings of this study highlight the potential for smartphone-based gait analysis in a rehabilitation setting. Future studies are required to further demonstrate the use of smartphone-based gait assessment in clinical and rehabilitation hospital settings.

## Figures and Tables

**Figure 1 sensors-24-03594-f001:**
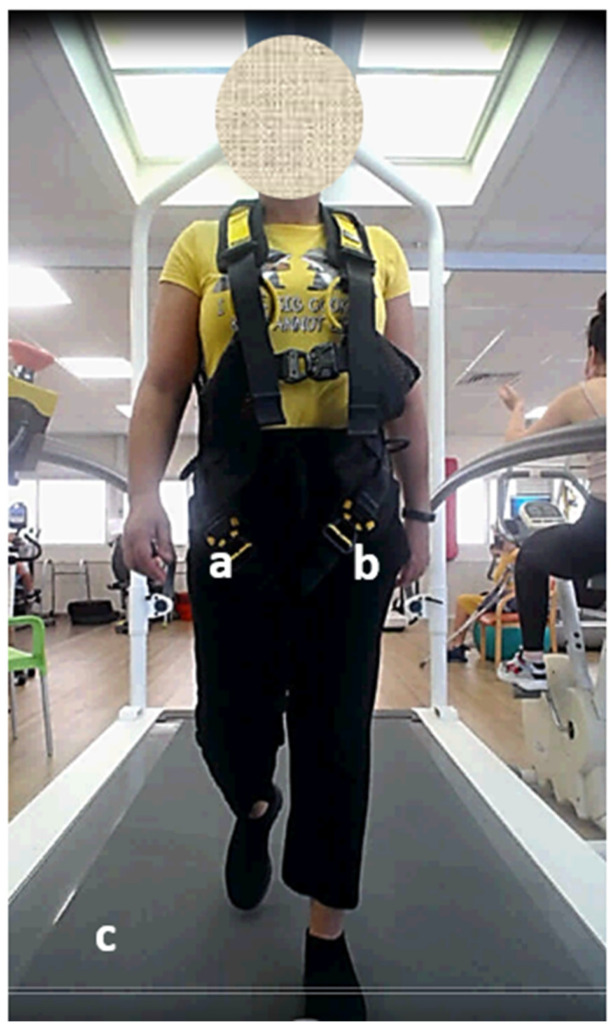
Description of data collection procedure. During each walking session, three devices concurrently recorded the participants’ spatiotemporal gait parameters: (**a**) A phone with the app, placed inside the pocket of the impaired leg; (**b**) A phone with the app, placed inside the pocket of the opposite leg; (**c**) The C-Mill treadmill. Each participant was required to wear a standard safety harness. The use of handrails was permitted.

**Figure 2 sensors-24-03594-f002:**
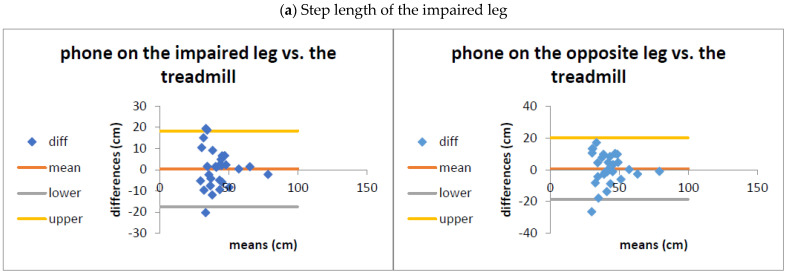
Bland–Altman plots: limits of agreement between the OneStep smartphone application and the treadmill, as measured on each leg: (**a**,**b**) Step length of the impaired/opposite leg; (**c**,**d**) Swing phase of the impaired/opposite leg; (**e**,**f**) Stance phase of the impaired/opposite leg; and (**g**,**h**) Single limb support of the impaired/opposite leg. The yellow and gray lines indicate the upper and lower 95% limits of agreement (±1.96 SD of the bias).

**Table 1 sensors-24-03594-t001:** Characteristics of the study participants (*n* = 28).

Variable	Category	Mean ± SD/n (%)
Age		42.4 ± 15.0
Sex	Male	17 (61)
	Female	11 (39)
BMI (kg/m^2^)	<18	2 (7)
	18–25	13 (46)
	>25	13 (46)
Affected leg	Left	18 (64)
	Right	10 (36)
Type of disability	Fracture/s of the lower extremity	14 (50)
	CVA/TBI (Hemiparesis)	8 (29)
	Chronic pain in the lower extremity	6 (21)
BPI		
Pain severity (0–10)		3.9 ± 2.6
Pain interference (0–10)		4.2 ± 2.6
LEFS (0–80)		30.3 ± 11.4

Abbreviations: BMI—Body Mass Index; BPI—Brief Pain Inventory; CVA—Cerebral Vascular Accident; LEFS—Lower Extremity Functional Scale; SD—Standard Deviation; TBI—Traumatic Brain Injury.

**Table 2 sensors-24-03594-t002:** Mean (standard deviation) of gait parameters, as measured by each of the three assessment tools.

Variable (Unit of Measurement)	Measured by the Instrumented Treadmill or Phone Placed on a Leg	Mean ± SD
Cadence (Steps per minute)	Opposite Leg	79.69 ± 22.07
	Impaired Leg	79.63 ± 22.05
	Treadmill	79.65 ± 22.00
Gait speed (Meters per second)	Opposite Leg	0.60 ± 0.33
	Impaired Leg	0.60 ± 0.34
	Treadmill	0.57 ± 0.34
Stride length (cm)	Opposite Leg	85.79 ± 21.56
	Impaired Leg	85.85 ± 20.93
	Treadmill	82.32 ± 23.46
Step length of the opposite leg (cm)	Opposite Leg	43.04 ± 10.43
	Impaired Leg	43.42 ± 10.43
	Treadmill	40.32 ± 12.13
Step length of the impaired leg (cm)	Opposite Leg	42.75 ± 11.80
	Impaired Leg	42.44 ± 11.38
	Treadmill	42.01 ± 12.11
Swing phase of the opposite leg (% gait cycle)	Opposite Leg	30.72 ± 4.61
	Impaired Leg	30.05 ± 4.24
	Treadmill	29.87 ± 5.86
Swing phase of the impaired leg (% gait cycle)	Opposite Leg	31.71 ± 4.47
	Impaired Leg	31.90 ± 4.40
	Treadmill	32.12 ± 6.11
Stance phase of the opposite leg (% gait cycle)	Opposite Leg	69.28 ± 4.61
	Impaired Leg	69.95 ± 4.24
	Treadmill	70.13 ± 5.86
Stance phase of the impaired leg (% gait cycle)	Opposite Leg	68.29 ± 4.47
	Impaired Leg	68.10 ± 4.40
	Treadmill	67.88 ± 6.11
Single limb support of the opposite leg (% stance phase)	Opposite Leg	31.71 ± 4.47
	Impaired Leg	31.90 ± 4.40
	Treadmill	32.12 ± 6.11
Single limb support of the impaired leg (% stance phase)	Opposite Leg	30.72 ± 4.61
	Impaired Leg	30.05 ± 4.24
	Treadmill	29.87 ± 5.86
Double limb support (% stance phase)	Opposite Leg	37.57 ± 6.60
	Impaired Leg	38.05 ± 6.79
	Treadmill	38.01 ± 7.71

Abbreviations: SD—Standard Deviation.

**Table 3 sensors-24-03594-t003:** Reliability tests of gait parameters between the two smartphones, one placed on the impaired leg and one placed on the opposite leg.

Variable (Unit of Measurement)	Pearson *r*	Mean Difference	*p*-Value ^a^	Cohen’s d	ICC Consistency (95% CI)
Cadence (Steps per minute)	1	0.06	0.08	0.29	1
Gait speed (Meters per second)	0.992	0.00	0.84	−0.02	0.992 (0.982, 0.996)
Stride length (cm)	0.935	−0.06	0.85	−0.01	0.934 (0.864, 0.969)
Step length of the opposite leg (cm)	0.905	−0.37	0.66	−0.08	0.905 (0.805, 0.955)
Step length of the impaired leg (cm)	0.911	0.31	0.85	0.06	0.911 (0.816, 0.958)
Swing phase of the opposite leg (% gait cycle)	0.910	0.67	0.08	0.35	0.907 (0.810, 0.956)
Swing phase of the impaired leg (% gait cycle)	0.888	−0.19	0.48	−0.09	0.888 (0.773, 0.947)
Stance phase of the opposite leg (% gait cycle)	0.910	−0.67	0.08	−0.35	0.907 (0.810, 0.956)
Stance phase of the impaired leg (% gait cycle)	0.888	0.19	0.48	0.09	0.888 (0.773, 0.947)
Single limb support of the opposite leg (% stance phase)	0.888	−0.19	0.48	−0.09	0.888 (0.773, 0.947)
Single limb support of the impaired leg (% stance phase)	0.910	0.67	0.08	0.35	0.907 (0.810, 0.956)
Double limb support (% stance phase)	0.935	−0.48	0.21	−0.20	0.935 (0.865, 0.969)

Abbreviations: CI—confidence interval; ICC—intra-class correlation coefficient. Notes: ^a^ Wilcoxon signed-rank test.

**Table 4 sensors-24-03594-t004:** Validity tests of gait parameters between the two smartphones and the treadmill.

Variable (Unit of Measurement)	Smartphone Location	Pearson *r*	Mean Difference	*p*-Value ^a^	Cohen’s d	ICC Consistency (95% CI)
Cadence (Steps per minute)	Opposite	1	0.04	0.28	0.17	1
	Impaired	1	−0.02	0.21	−0.11	1
Gait speed (Meters per second)	Opposite	0.966	0.02	0.15	0.28	0.966 (0.928, 0.984)
	Impaired	0.968	0.03	0.22	0.30	0.968 (0.933, 0.985)
Stride length (cm)	Opposite	0.806	3.46	0.15	0.25	0.803 (0.618, 0.904)
	Impaired	0.851	3.53	0.21	0.29	0.846 (0.694, 0.925)
Step length of the opposite leg (cm)	Opposite	0.791	2.72	0.04 †	0.36	0.782 (0.581, 0.893)
	Impaired	0.821	3.09	0.03 †	0.45	0.811 (0.633, 0.908)
Step length of the impaired leg (cm)	Opposite	0.654	0.74	0.46	0.07	0.653 (0.377, 0.823)
	Impaired	0.699	0.43	0.98	0.05	0.698 (0.445, 0.848)
Swing phase of the opposite leg (% gait cycle)	Opposite	0.879	0.85	0.02 †	0.30	0.854 (0.710, 0.930)
	Impaired	0.830	0.18	0.27	0.05	0.788 (0.593, 0.896)
Swing phase of the impaired leg (% gait cycle)	Opposite	0.836	−0.41	0.98	−0.12	0.797 (0.608, 0.901)
	Impaired	0.825	−0.22	0.45	−0.06	0.782 (0.583, 0.893)
Stance phase of the opposite leg (% gait cycle)	Opposite	0.879	−0.85	0.02 †	−0.30	0.854 (0.710, 0.930)
	Impaired	0.830	−0.18	0.27	−0.05	0.788 (0.593, 0.896)
Stance phase of the impaired leg (% gait cycle)	Opposite	0.836	0.41	0.98	0.12	0.797 (0.608, 0.901)
	Impaired	0.825	0.22	0.45	0.06	0.782 (0.583, 0.893)
Single limb support of the opposite leg (% stance phase)	Opposite	0.836	−0.41	0.98	−0.12	0.797 (0.608, 0.901)
	Impaired	0.825	−0.22	0.45	−0.06	0.782 (0.583, 0.893)
Single limb support of the impaired leg (% stance phase)	Opposite	0.879	0.85	0.02 †	0.30	0.854 (0.710, 0.930)
	Impaired	0.830	0.18	0.27	0.05	0.788 (0.593, 0.896)
Double limb support (% stance phase)	Opposite	0.815	−0.44	0.49	−0.10	0.805 (0.621, 0.905)
	Impaired	0.870	0.04	0.80	0.01	0.863 (0.726, 0.934)

Abbreviations: CI—confidence interval; ICC—intra-class correlation coefficient. Notes: ^a^ Wilcoxon signed-rank test; † Statically significant at *p*-value < 0.05.

## Data Availability

The datasets used and analyzed during the current study are available from the corresponding author on reasonable request.

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
