# Peer review of "The Reliability and Validity of the OneStep Smartphone Application for Gait Analysis among Patients Undergoing Rehabilitation for Unilateral Lower Limb Disability"

_sensors, 2024, doi:10.3390/s24113594_

Round 1
Reviewer 1 Report
Comments and Suggestions for Authors
The manuscript introduces the evaluation results of an easy-to-use and reliable OneStep smartphone applications. By comparing to the C-Mill-VR+treadmill, the correlations of varying gait parameters between the phones and the treadmill are mostly above 0.8 in patients with unilateral lower limb disability. The study designs valid and reasonable experiments to collect data and make the comparison. The results are convincing and the discussion is insightful. However, I have some concerns and suggestions regarding the details of the method.
--To make the paper self-contained, the methods of estimating different gait parameters are needed to be presented and described in details.
--Some related references for gait analysis and gait parameter estimation are recommended.
[1] Acceleration and electromyography (EMG) pattern analysis for children with cerebral palsy
[2] IMU-based joint angle measurement for gait analysis
Reviewer 2 Report
Comments and Suggestions for Authors
1. Why did the author choose to prove the reliability and validity by comparing mobile phone application with treadmills?
2. Why wasn't the duration of treadmill usage standardized for each participant in the experiment?
3. I think it is better to add a gold standard to establish the accuracy and reliability of measuring gait parameters using both mobile phone application and treadmills.
4. How does this study demonstrate the conclusion that " this study adds to previous ones [22, 23, 31] by demonstrating that among the examined population, slower gait speeds and shorter strides do not jeopardize the app's accuracy in assessing spatiotemporal gait parameters." From the methods, the study does not directly give standards for slower walking speeds and shorter strides.
5. Is it necessary for the experiment to include data from healthy individuals to demonstrate differences between data collected by mobile apps or treadmills for both healthy individuals and patients with leg injuries?
Reviewer 3 Report
Comments and Suggestions for Authors
In this study, the Authors assessed the validity of a mobile phone app in monitoring and evaluating spatio-temporal gait parameters in subjects with unilateral lower limb disability. For this purpose, they used two mobile phones, one placed on the impaired leg and the other placed on the opposite leg, and compared parameters obtained from both phones to those obtained by a reference instrumented treadmill. The Authors found a good agreement between the two phones, thus validating the repeatability (or precision) of the mobile phone application, and between phone and threadmill measurements, thus validating the accuracy of the proposed solution.
The work is really interesting, paving the way to out-of-lab gait monitoring and assessment in rehabilitation protocols. However, a few points need to be addressed:
1. Does the app use raw or compensated signals? It is well known that low-cost inertial sensors, like those on smartphones, suffer from errors due to drift, nonlinearity, temperature, axes misalignment, etc… However, the use of combined accelerometers, gyroscopes and magnetometers can help correct errors, for example, using information fusion techniques like Kalman filters, etc. I suspect that using compensated signals may help improve accuracies.
2. In the Methods section, the Authors refer to “the p value for a test for difference”. In Table 4 the Wilcoxon signed rank test is reported. It shoud be reported in the methods section, too, explaining the motivation for this choice (e.g.: did the Authors perform a preliminary test on the distribution of data?)
Comments on the Quality of English LanguageI have no comments except the following.
line 199: “For one”. Maybe “First” could sound better.
Round 2
Reviewer 2 Report
Comments and Suggestions for Authors
No further comments.